# Design of an Ultrasound Transceiver ASIC with a Switching-Artifact Reduction Technique for 3D Carotid Artery Imaging

**DOI:** 10.3390/s21010150

**Published:** 2020-12-29

**Authors:** Taehoon Kim, Fabian Fool, Djalma Simoes dos Santos, Zu-Yao Chang, Emile Noothout, Hendrik J. Vos, Johan G. Bosch, Martin D. Verweij, Nico de Jong, Michiel A. P. Pertijs

**Affiliations:** 1Electronic Instrumentation Laboratory, Delft University of Technology, 2628 CD Delft, The Netherlands; Z.Y.Chang@tudelft.nl (Z.-Y.C.); M.A.P.Pertijs@tudelft.nl (M.A.P.P.); 2Laboratory of Medical Imaging, Department of Imaging Physics, Delft University of Technology, 2628 CJ Delft, The Netherlands; F.Fool@tudelft.nl (F.F.); D.SimoesdosSantos@tudelft.nl (D.S.d.S.); E.C.Noothout@tudelft.nl (E.N.); h.vos@erasmusmc.nl (H.J.V.); M.D.Verweij@tudelft.nl (M.D.V.); Nicolaas.deJong@tudelft.nl (N.d.J.); 3Department Biomedical Engineering, Thoraxcenter, Erasmus Medical Center, 3015 GD Rotterdam, The Netherlands; j.bosch@erasmusmc.nl

**Keywords:** 3D ultrasound imaging, high-voltage (HV) switches, matrix transducers, ultrasound application-specific integrated circuit (ASIC), clock feedthrough, charge injection

## Abstract

This paper presents an ultrasound transceiver application-specific integrated circuit (ASIC) directly integrated with an array of 12 × 80 piezoelectric transducer elements to enable next-generation ultrasound probes for 3D carotid artery imaging. The ASIC, implemented in a 0.18 µm high-voltage Bipolar-CMOS-DMOS (HV BCD) process, adopted a programmable switch matrix that allowed selected transducer elements in each row to be connected to a transmit and receive channel of an imaging system. This made the probe operate like an electronically translatable linear array, allowing large-aperture matrix arrays to be interfaced with a manageable number of system channels. This paper presents a second-generation ASIC that employed an improved switch design to minimize clock feedthrough and charge-injection effects of high-voltage metal–oxide–semiconductor field-effect transistors (HV MOSFETs), which in the first-generation ASIC caused parasitic transmissions and associated imaging artifacts. The proposed switch controller, implemented with cascaded non-overlapping clock generators, generated control signals with improved timing to mitigate the effects of these non-idealities. Both simulation results and electrical measurements showed a 20 dB reduction of the switching artifacts. In addition, an acoustic pulse-echo measurement successfully demonstrated a 20 dB reduction of imaging artifacts.

## 1. Introduction

Real-time 3D ultrasound imaging is an essential technique for the accurate assessment of carotid artery disease by measuring blood flow, plaque deformation, and pulse wave velocity [1,2,3,4]. To realize this, the next generation of ultrasound probes require matrix transducer arrays with thousands of elements to cover a sufficiently large aperture (>400 mm^2^). It is possible to build a matrix array with such a high number of elements, but making electrical connections to all the elements is highly challenging since conventional imaging systems have a limited number of channels.

This issue can be addressed by integrating application-specific integrated circuits (ASICs) into the probe to reduce the number of electrical connections via cables to the imaging system. Various approaches to interface such large-aperture matrix transducer arrays using a reduced number of cables have been reported, such as receive sub-array beamforming [5,6,7,8], programmable high-voltage (HV) pulsers [9,10,11,12,13,14], switch matrices [15,16], row-by-row scanning schemes [12,14], and row- or column-parallel connection schemes [13].

We previously reported a first-generation transceiver ASIC with a row-level architecture for both receive (RX) and transmit (TX) channels [17]. This ASIC consists of 24 × 40 element-level HV switches and control logic that allows selected transducer elements in each row to be connected to a TX and RX channel of an imaging system. The element-level circuits are laid out in a pitch-matched fashion and connect directly via bondpads to transducer elements stacked on top of the chip. Although the functionality of this ASIC has been successfully demonstrated in a 3D imaging experiment, noticeable imaging artifacts associated with parasitic transmissions generated by HV switch actuation at the TX to RX transition and vice versa were found. In particular, thousands of switches actuate simultaneously in a full-aperture selection for plane wave transmission, making this problem more serious. The parasitic transmissions originate from non-idealities of the HV MOSFETs in the switch circuits, i.e., clock feedthrough and charge injection, generating switching transients on the transducer elements.

Charge injection and clock feedthrough in low-voltage analog complementary metal–oxide–semiconductor (CMOS) circuits have been discussed extensively in the literature [18], and several techniques to cancel their effects have been reported, such as dummy transistor compensation [19,20] and differential clock feedthrough attenuation [21]. However, these techniques are not directly applicable in this design because the HV laterally diffused metal oxide semiconductor (LDMOS) transistors used for HV switches are much larger than low-voltage transistors. As a result, these solutions, which rely on additional transistors, would make die area occupied by the element-level switch circuit too large in comparison to the size of the transducer elements. Recently, one approach, linearized control of the gate voltage of a HV MOSFET, has been reported to address this issue in an ASIC design with large aperture transducer arrays for ultrasound imaging applications [22].

In this work, as an alternative, we propose a second-generation transceiver ASIC focusing on minimizing the effects due to non-idealities of the HV MOSFETs, the root cause of the mentioned imaging artifacts. To achieve this, we propose a new HV switch controller that generates control signals for the HV switches with improved timing to alleviate the effects of their non-idealities. In this design, a current discharging path from the transducer element to the ground is created at the TX and RX switching moments. Consequently, this leads to significantly reduced peak-to-peak amplitudes of the switching transients.

This paper is structured as follows. Section 2 describes the materials and methods of this work, which comprise a system overview in Section 2.1, the architecture of the designed ASIC in Section 2.2, the implementation of the element-level switch circuit and an analysis of the associated switching transients due to non-idealities of the HV MOSFETs in Section 2.3, the design of a switch controller with a switching-artifact reduction technique in Section 2.4, and its validation in simulation in Section 2.5. Section 3 describes the measurement results and discussion, explaining the experimental prototype used to validate the design in Section 3.1, presenting its electrical verification in Section 3.2, and showing reduced imaging artifacts in Section 3.3. Finally, the paper is concluded in Section 4.

## 2. Materials and Methods

### 2.1. System Overview

The size of the matrix transducer to cover an aperture sufficiently large enough for carotid artery imaging is 36 × 12 mm^2^. To facilitate the realization of a full-sized array, we designed the second-generation ASIC via a MLM (multi-layer mask) full-wafer fabrication process. Compared to our previous work [17], the row pitch was increased from 150 to 300 µm while keeping the center frequency as it was. Thus, the number of rows in the ASIC was reduced from 24 to 12 so that the density of the wire bonds that electrically connect the ASIC to a printed circuit board (PCB) was also significantly reduced. This modification of physical dimensions enabled us to wire bond more reliably, which is one of the main challenges of the first-generation ASIC. On the other hand, the number of columns per ASIC increased from 40 to 80, which allowed us to implement the targeted aperture by tiling 1 × 10 ASICs, as shown in Figure 1a. This increased number of columns provided an important additional advantage compared to the previous work with 2 × 10 ASICs tiled, since wire-bond connections only have to be made on one side of the tiled array.

As shown in Figure 1b, each of the ASICs had a die size of 3.6 × 12.8 mm^2^, and interfaced with an array of 12 × 80 (rows × columns) transducer elements through element-level and row-level circuits. Each pitch-matched element-level circuit was connected to the corresponding transducer element via a bondpad. Bondpads on the periphery of each ASIC were arranged in two columns. The inner column connected row-level TX and RX signals to an imaging system. Bondpads of the outer column made connections for digital control signals as well as analog and digital power supplies.

In the same way as the first-generation ASIC, the piezoelectric layer (PZT)-on-CMOS integration scheme illustrated in Figure 2 [23] was adopted to build the transducer array on top of the ASICs. To electrically connect the circuitry on the ASIC to the transducer elements, we formed gold bumps on top of the ASIC bondpads using a wire-bonding tool. This was done in a matrix pattern with the same pitch as the transducer array. A nonconductive epoxy buffer layer was then deposited, filling the gaps between the gold. After this, the buffer layer was ground down to expose the gold, thus providing reliable electrical contacts for the transducer elements. On top of the epoxy layer, the acoustic stack consisting of a PZT and a matching layer was constructed. The conductive glue layer created the electrical connection between contacts and the electrode on the back-side of the piezoelectric ceramic. After that, the stack was diced into the desired 150 × 300 µm pitch array pattern using a diamond saw. The dicing kerfs were air-filled to minimize the crosstalk between elements. Finally, the array was covered with an aluminum foil that formed the elements’ common ground electrode.

### 2.2. ASIC Architecture

Figure 3 shows the top-level architecture of the second-generation ASIC. The 80 elements in each row of the matrix shared a row-level RX and TX bus to reduce channel count by a factor of 80. Even with 1 × 10 tiled ASICs, the total number of RX and TX channels was 240, which could be managed by an imaging system. Each transducer element was connected to the RX bus or the TX bus through a programmable element-level circuit. The RX bus was associated with a shared row-level circuit, which amplified the received echo signals, filtered out-of-band noise, and drove the connection to the imaging system.

Ahead of starting successive transmit/receive cycles, we needed to program the ASIC. Row-level logic was used to select one of the various element-selection modes supported by the ASIC to implement a specific imaging scheme and set the gain of the row-level circuit. This logic was programmed through a vertical shift registers (VSR) that was loaded during the programming phase. Each element was associated with logic allowing us to change rapidly between transmit/receive cycles the selection of active elements for a specific transmit and/or receive cycle. This element-level logic was pre-loaded through a horizontal shift register (HSR) during the programming phase. The switch controller received low-voltage digital signals (Φ_TX_ and Φ_RX_) from a field-programmable gain array (FPGA) and generated 5-V control signals (Φ_TX,PZT_, Φ_RX,PZT_, Φ_RX,GND_, and Φ_RX,LNA_), according to the pre-loaded data, to drive the element-level switch circuit. In this design, the switch controller was implemented using a non-overlapping clock generator to minimize the non-idealities of HV devices introduced above in the switch circuit.

### 2.3. Element-Level Switch Circuit and Non-Idealities

Figure 4 shows the circuit diagram of the element-level switches implemented in the first-generation ASIC. The HV switches were needed to enable HV pulse transmission while keeping a more compact implementation and less power consumption. The control signals were generated by simple combinational logic from the TX/RX phases (Φ_TX_ and Φ_RX_) and three enable bits (ELE_EN[i, j], TX_EN[i, j], and RX_EN[i, j]), which were pre-loaded via the HSR in the programming phase. The ELE_EN bit determined whether an element was enabled (high) or disabled (low). If a certain element was disabled, it was excluded from participating in imaging and connected to the ground via transistors M5 and M6. This helped to prevent the signal of disabled elements from coupling to the RX bus and capacitive coupling from the TX bus to the disabled elements in the TX phase Φ_TX_. For the enabled elements, the remaining two bits, TX_EN and RX_EN, determined whether they participate in transmission and/or reception.

During the TX phase Φ_TX_, the row-level TX bus, TX[i] was connected to the element if ELE_EN[i, j] and TX_EN[i, j] were high. Two back-to-back n-type HV LDMOS transistors (M1 and M2), driven by Φ_TX, PZT_, allowed an external imaging system to send a unipolar pulse with a peak value up to 65 V. To turn on M1 and M2, we charged a bootstrap capacitor C1 connected between their source and gate through M3 by making Φ_TX,SW_ high at the beginning of Φ_TX_. Soon after that, Φ_TX,SW_ went low to turn off M3, keeping M1 and M2 on and allowing them to swing up with the transmit HV pulses on the TX bus. To provide a sufficiently high gate voltage to turn on M1 and M2, we gave the bootstrap capacitor C1 a relatively large value of 7.2 pF in this design. At the end of Φ_TX_, C1 was discharged by pulling down the source of M3 so that M1 and M2 were turned off. During the RX phase Φ_RX_, transistors M4 and M6 connected the element to the RX bus if ELE_EN[i, j] and RX_EN[i, j] were high.

As described earlier in Section 1, an important consideration in this design is minimizing the impact of non-idealities, clock feedthrough, and charge injection of the HV switches, which generate switching glitches on the transducer element at the TX to RX transition and vice versa. This problem eventually leads to visible imaging artifacts. Figure 5a shows simulated waveforms of the control signals driving the element-level switch transistors. Even without a transmitted pulse, noticeable switching glitches appear on the element, as shown in Figure 5b. The peak amplitudes of these undesired glitches at the RX to TX transition and vice versa were 2.19 V and −0.64 V, respectively. This amplitude, in the volt range, led to visible imaging artifacts, even when applying a high-voltage transmit pulse with an amplitude of tens of volts.

### 2.4. Proposed Switch Controller

To minimize the switching transients discussed in the previous section, we proposed a new timing diagram, shown in Figure 6a, of the switch control signals. At the transition from RX to TX, the RX bus was first disconnected from the element by making Φ_RX, LNA_ low while Φ_RX, PZT_ remained high. Subsequently, M5 was turned on by making Φ_RX, GND_ high; thus, a low-impedance discharging path from the element to ground was created before actuating M1 and M2 associated with the HV TX switch. Thus, when Φ_TX, PZT_ was made high, the clock feedthrough-induced current could flow to the ground through transistor M5 instead of to the element. Finally, when Φ_RX, PZT_ was made low, the charge injection error from M4 could be absorbed to both ground and the low-impedance output of the pulser. This switching sequence reduced the peak amplitude of the switching transients significantly at the transition of RX to TX. Although M6, a 5 V device, was switching without a discharging path to the ground, the associated transient was not critical because M6 was much smaller than the high-voltage devices.

Conversely, at the transition of TX to RX, transistor M4 was activated first to create a discharging path again from the element to the ground. At the transition of Φ_TX, PZT_, a significant error could be generated due to charge injection related to M1 and M2. However, the resulting voltage transient on the element was minimized because the associated charge could flow via M4 and M5 to the ground.

Figure 6b shows the switch controller’s design with cascaded non-overlapping clock generators to generate the control signals presented in Figure 6a. Each stage employed delay unit cells in which metal–oxide–semiconductor (MOS) capacitors were used to make sufficient delay (>100 ns) between the signals.

### 2.5. Validation in Simulation

The proposed switch controller was simulated to validate the reduction of the switching transients. Figure 7 shows the simulation results of the control signals generated by the controller, the voltage on the element, and the current flowing through M5 in Figure 4 at the switching moments for the TX to RX transition and vice versa. Figure 7a shows the transition of the signals at the switching moment from RX to TX, with the peak-to-peak amplitude of switching transients being 139.2 mV, as shown in Figure 7b. Figure 7c shows that the current flows through M5 at the switching moments of high-voltage transistors M1, M2, and M4, resulting in the reduction of the switching transients. Figure 7d shows waveforms of the control signals at the transition from TX to RX, with the peak-to-peak amplitude being 207.3 mV, as shown in Figure 7e. In particular, the largest peak value of −118.8 mV, associated with the falling edge of Φ_TX, PZT_, was caused by the charge injection effect of M1 and M2. This amplitude was significantly reduced because the charge-injection-induced current was also absorbed into the ground via M5, as is confirmed in Figure 7f. Consequently, compared to 2.19 V in the first-generation ASIC simulation, the peak-to-peak amplitude of the switching transients with the newly designed switch controller was reduced by around 20 dB.

## 3. Results and Discussion

### 3.1. Experimental Prototype

The ASIC was fabricated in a 0.18-µm HVBCD process. Figure 8a,b shows a photograph of a bare die and a plot of the layout of the element-level TX and RX circuits, respectively. The area of the element-level circuit was matched to the 150 × 300 µm^2^ transducer element size. In the prototype shown in Figure 8c, four ASICs were mounted on a test PCB and wire-bonded for RX and TX channels as well as for power and control signals. A piezo-electric transducer array was built on top of the ASICs using the process described in [23]. The prototype was covered by a ground foil, the common ground electrode of the transducer elements, and by a moisture protection layer.

### 3.2. Electrical Verification

In the electrical characterization, an ASIC die without transducer array was used, on which selected transducer pads were wire-bonded to the test board to observe the switching transients. Without applying a high-voltage TX pulse, residual switching transients were visible, as shown in Figure 9. Their peak-to-peak amplitudes at the TX to RX transition and vice versa were 100 mV_PP_ and 30 mV_PP_, respectively. Compared to the simulation results with the first-generation ASIC shown in Figure 5, the peak-to-peak values were reduced by factors of 6 and 70. Compared to the simulation results shown in Figure 7, the measured peak-to-peak amplitudes were a factor of 2 and 5 lower. This was likely due to the effects of the wire-bonded connection, the bondpad on the PCB, and the probe capacitance in the test setup, which resulted in a higher capacitance, and hence a smaller transient voltage than in the simulation. Taking this loading effect into account, the results were consistent with the expected reduction of 20 dB derived in Section 2.5.

### 3.3. Improvement in Clock Feedthrough and Charge Injection-Induced Imaging Artifact

The test-bench for acoustic experiments to validate the reduced switching artifacts is shown in Figure 10. A Verasonics V1 imaging system (Verasonics, Kirkland, WA, USA) was used to generate the HV TX signals. To interface the TX and RX signals between the Verasonics and the test PCB, we designed a motherboard PCB. On the transmit paths, a matching circuit was implemented to guarantee a unipolar excitation pulse. The received signals were buffered with unity-gain operational amplifiers on the motherboard before being fed into the Verasonics. The motherboard also included low-dropout regulators (LDOs) to generate supply voltages for analog, digital, and 5 V circuitry. Moreover, the board contained digital buffers to transfer the signals from the field-programmable gate array (FPGA) board. These digital signals consisted of clock and data for programming the ASICs and their control signals.

Pulse-echo measurements were performed to obtain imaging results using a quartz plate as a reflector located at a depth of 13 mm. The TX signal was a unipolar half-cycle 7.5 MHz pulse with a peak value of 1 V. All elements were enabled to send a plane wave on the full-aperture in TX, and echo signals were received in a column-by-column fashion. The reason for using a relatively small 1 V pulse was to better bring out the switching artifacts compared to the 60 V pulse used in normal operation. To record the switching glitches at the transitions of both TX to RX and vice versa, we applied two successive RX phases, as shown in Figure 11. During the first RX phase, the echo signal from the transmitted 1 V pulse and the switching transients from the first transition from TX to RX (the first rising edge) can be recorded. In the second cycle, the switching transients from the first transition from RX to TX (the first falling edge) and from the second transition from TX to RX (the second rising edge) were recorded.

Figure 12 demonstrates how the switching transients affected the image and how much imaging artifacts were improved with the proposed switch controller circuit. Figure 12a,b show the RF signal for a single channel captured by the Verasonics and the imaging result, respectively, with the first-generation ASIC. As can be seen in Figure 12a, the peak-to-peak amplitude of the RX to TX switching transients located at a depth of 38 mm was considerably larger than the echo signal from the transmitted 1 V pulse. Even if the largest pulse amplitude supported by the ASIC of 65 V would be used, these switching transients would give significant artifacts in the image. On the other hand, the amplitude of the RX to TX switching transients was reduced by a factor of 20 dB in Figure 12c,d for the second-generation ASIC. This reduced amplitude corresponded to a parasitic transmitted pulse with an amplitude of only 0.3 V. If the maximum pulse amplitude of 65 V were to be used, the echo signals from the switching would be 46.7 dB smaller than the echoes from the intended pulse, making their impact on the image much less significant. This result demonstrated that the switching artifact reduction scheme with the proposed switch controller improved image quality significantly.

## 4. Conclusions

We present an ultrasound transceiver ASIC design with a switching artifact reduction technique, designed for 3D volumetric imaging of the carotid artery. We have previously reported a first-generation ASIC design [17] that demonstrated the 3D imaging capability; however, we observed noticeable ghost echoes in the experiment. We have identified that this issue originates from clock feedthrough and charge injection of the HV MOSFETs in the element-level switch circuits, generating switching transients on the transducer elements that lead to parasitic pulse transmission. The improved switch controller proposed in this paper generates control signals of HV switches such that a discharge path to ground is created at the critical switching moments, strongly reducing the transients on the transducer element. Measured results demonstrate a 20 dB reduction in imaging artifacts compared to our previous ASIC.

## Figures and Tables

**Figure 1 sensors-21-00150-f001:**
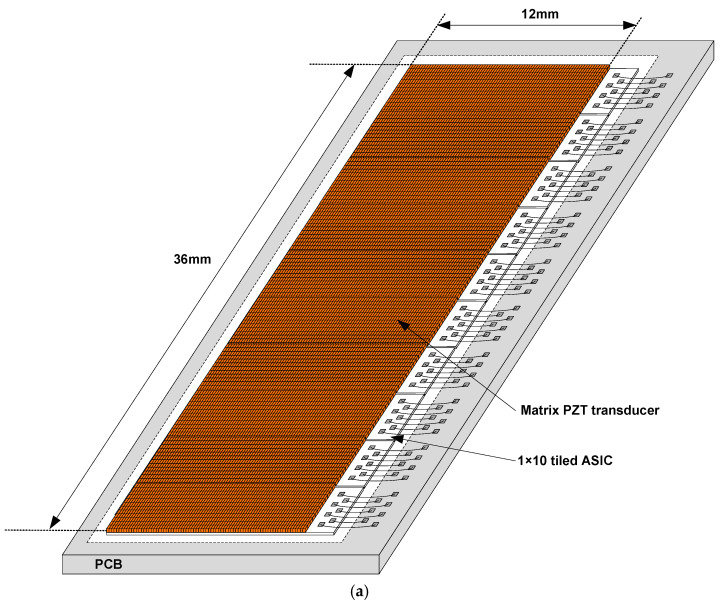
(**a**) Overview of the proposed matrix transducer on tiled application-specific integrated circuits (ASICs). (**b**) Overview of a single tile consisting of 12 × 80 transducer elements on top of an ASIC with 12 × 80 element-level switches and row-level circuits.

**Figure 2 sensors-21-00150-f002:**
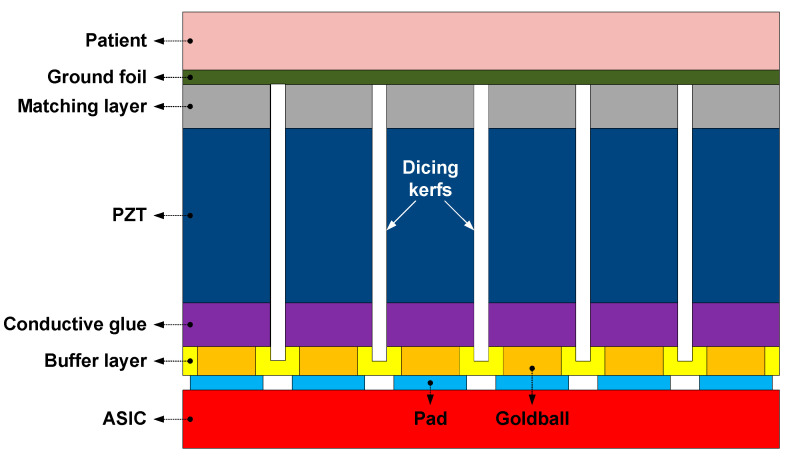
Cross-sectional view of the transducer array mounted on top of the ASIC.

**Figure 3 sensors-21-00150-f003:**
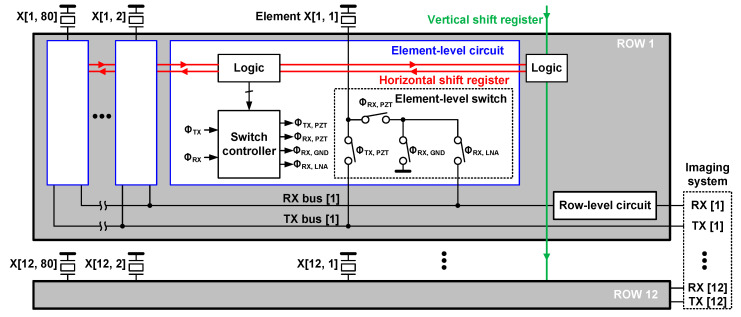
Block diagram of the second-generation ASIC.

**Figure 4 sensors-21-00150-f004:**
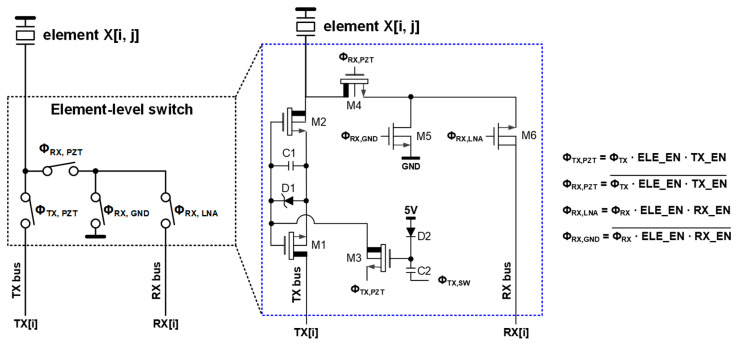
Simplified circuit diagram of the element-level switch and the combinational logic equation, implemented as the switch controller, proposed in [17].

**Figure 5 sensors-21-00150-f005:**
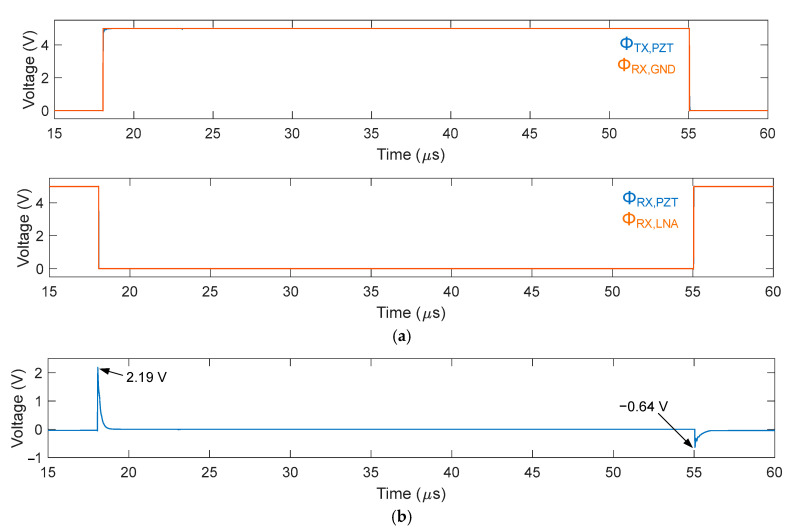
Simulated waveforms of the element-level circuit implemented in the first-generation ASIC (Figure 4): (**a**) control signals driving the element-level switch circuit and (**b**) the switching transients on the transducer element.

**Figure 6 sensors-21-00150-f006:**
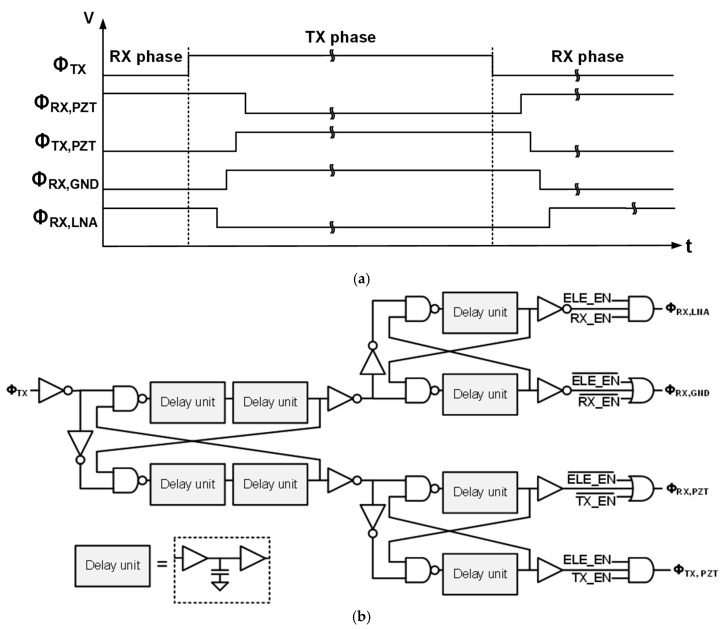
(**a**) The proposed timing diagram of the control signals to minimize the switching transients and (**b**) implementation of the switch controller to generate such signals.

**Figure 7 sensors-21-00150-f007:**
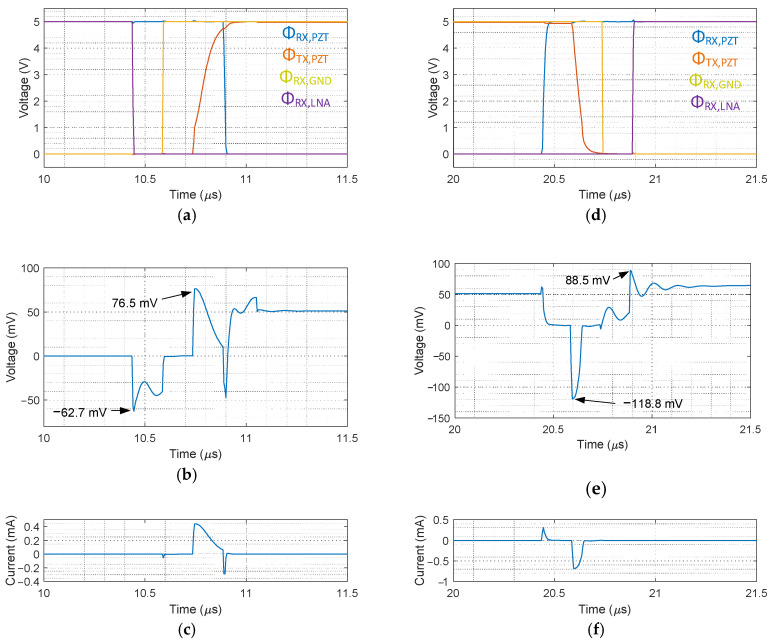
Simulated waveforms with the proposed switch controller at the transition of RX to TX (**a**–**c**) and vice versa (**d**–**f**). (**a**,**d**) Control signals driving the element-level switch circuit, (**b**,**e**) the switching transients on the transducer element, and (**c**,**f**) the drain current flowing via M5.

**Figure 8 sensors-21-00150-f008:**
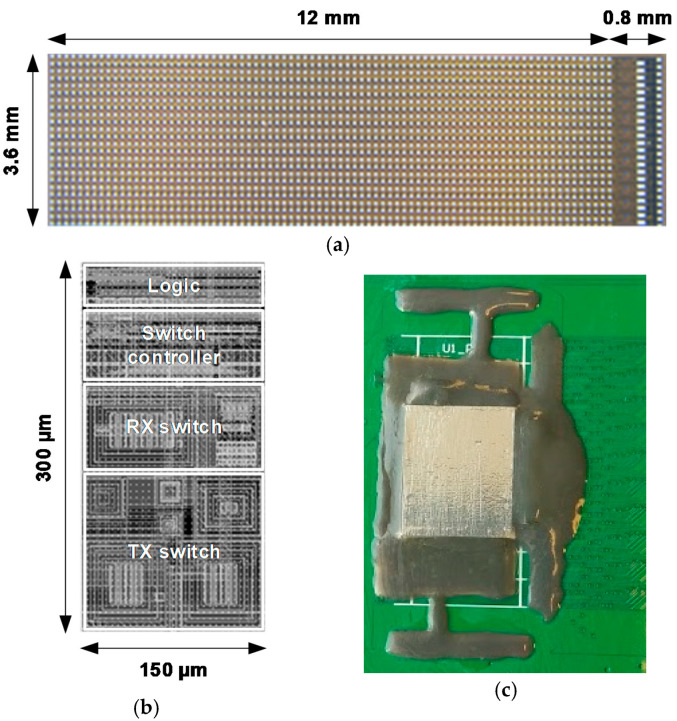
(**a**) Chip photograph of the ASIC, (**b**) layout of the pitch-matched element-level circuit, and (**c**) the test printed circuit board (PCB) with transducer array on top of the ASICs.

**Figure 9 sensors-21-00150-f009:**
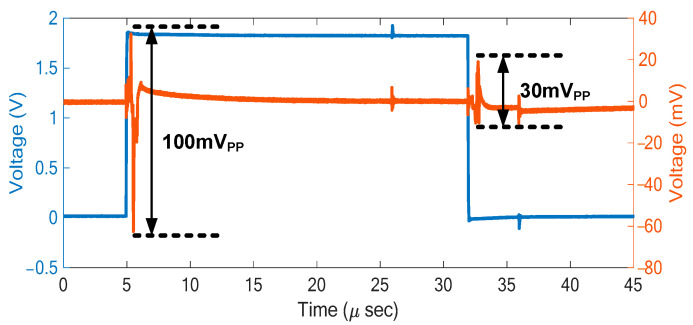
Measured waveforms of the RX phase (blue) and the switching transient observed on the element X [6, 41] (orange).

**Figure 10 sensors-21-00150-f010:**
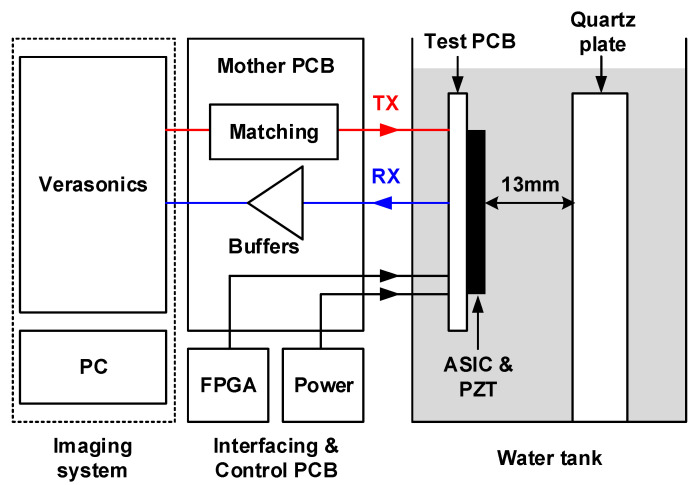
Overview of the measurement setup used for demonstrating the effect of the switching transients for the first- and the second-generation ASICs.

**Figure 11 sensors-21-00150-f011:**
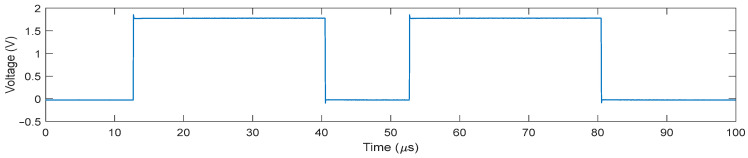
The waveform of the RX phase with two cycles used in the pulse-echo measurement to record the switching glitches at both TX to RX transition and vice versa.

**Figure 12 sensors-21-00150-f012:**
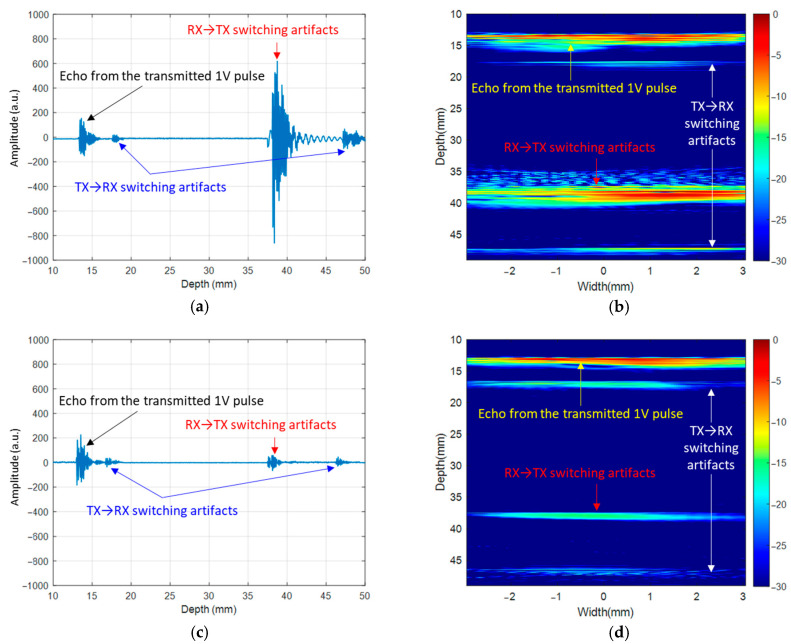
Comparison of switching transients for the first- and second-generation ASICs. (**a**) Time-domain plot of the received RX output data for a single channel and (**b**) B-Mode imaging result with the first-generation ASIC. (**c**) Time-domain plot for the second-generation ASIC with the same test conditions and (**d**) its imaging result.

## Data Availability

The data presented in this study are available on request from the corresponding author.

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
