# Peer review of "Design of an Ultrasound Transceiver ASIC with a Switching-Artifact Reduction Technique for 3D Carotid Artery Imaging"

_sensors, 2020, doi:10.3390/s21010150_

Round 1
Reviewer 1 Report
This manuscript entitled "Design of an Ultrasound Transceiver ASIC with a Switching-Artifact Reduction Technique for 3-D Carotid Artery Imaging", proposed a new switching algorithm to reduce the switching artifacts which degrade the ultrasound image quality.
In this manuscript, the architecture of the proposed ASIC is well described. However, the experimental result should be explained in more detail.
Please clarify how much of the impact the 20 dB noise reduction will have on the actual imaging.
In the experiment, the target was a quartz plate and its echo signal was not covered from the switching artifacts. In the actual use, the artifacts will cover the echoes from many reflectors.
Reviewer 2 Report
Dear authors,
your paper intitled "Design of an Ultrasound Transceiver ASIC 2 with aSwitching-Artifact Reduction Technique 3 for 3-D Carotid Artery Imaging" is really interesting.
The paper is well written, but in my opinion an overview on the cross section of ASIC is mandatory and it is to be added in the manuscript. Please add arepresentative figure, in which you can show layer by layer the construction/manufacturing of the sensor (from bottom to the patient).
Are there matching layers?
And about the backing?
Did you use a filling between the PZT squares?
There is no description of the material involved.
Take care.
